# Poly(ε-Caprolactone) Resorbable Auxetic Designed Knitted Scaffolds for Craniofacial Skeletal Muscle Regeneration

**DOI:** 10.3390/bioengineering7040134

**Published:** 2020-10-24

**Authors:** Monica V. Deshpande, Andre J. West, Susan H. Bernacki, Kun Luan, Martin W. King

**Affiliations:** 1Wilson College of Textiles, North Carolina State University, Raleigh, NC 27695, USA; mdeshpa3@ncsu.edu (M.V.D.); ajwest2@ncsu.edu (A.J.W.); kluan@ncsu.edu (K.L.); 2Joint Department of Biomedical Engineering, University of North Carolina, Chapel Hill & North Carolina State University, Raleigh, NC 27599, USA; shbernac@ncsu.edu; 3College of Textiles, Donghua University, Shanghai 201620, China

**Keywords:** auxetic design, biodegradable, craniofacial microsomia, knitted textile, muscle scaffold, negative Poisson’s ratio

## Abstract

Craniofacial microsomia is a congenital deformity caused by asymmetric development of the skull (cranium) and face before birth. Current treatments include corrective surgery and replacement of the deformed structure using autograft tissue, which results in donor site morbidity. An alternative therapy can be achieved by developing a resorbable scaffold for skeletal muscle regeneration which will help restore the symmetry and function of the facial muscles and reduce donor site morbidity. Two resorbable weft knitted scaffolds were fabricated using poly(ε-caprolactone) multifilament yarns with unique auxetic design structures possessing negative Poisson’s ratio (NPR). These scaffolds exhibit their NPR elasticity through an increase in total volume as well as no lateral narrowing when stretched longitudinally, which can provide orientated mechanical supports to the cell growth of skeletal muscle regeneration. These scaffolds were evaluated for the required physical properties, mechanical performance and biocompatibility by culturing them with neonatal human dermal fibroblasts so as to determine their cell metabolic activity, cell attachment and proliferation. This study can facilitate the understanding and engineering of textile-based scaffolds for tissues/organs. The work also paves a pathway to emerge the NPR textiles into tissue engineering, which has an extensive potential for biomedical end-uses.

## 1. Introduction

Craniofacial microsomia is a congenital deformity which affects approximately 1 in every 5600 children across the USA. It is characterized by improper growth of skull and face tissues on one or both sides of the face which creates a structural deficiency leading to clinical abnormalities like functional loss in hearing, swallowing, breathing and feeding, and hinders the child’s aesthetic development, such as facial expressions and appearance leading to social issues like acceptance [1,2]. The causes of the deformity are unknown, but it is thought to be the result of a genetic abnormality during embryo development [1]. The treatment involves a combination of four procedures, namely (i) reconstruction of the mandibular ramus, (ii) mandibular distraction and osteogenesis, (iii) bone grafting and (iv) orthognathic surgery, depending on the pathology and anatomy of the defect in the individual patient [3]. The corrective reconstructive surgery in young patients involves the repairing of deformed jawbones and facial structure followed by filling up the gap with a muscle graft, which has tensile and biological properties similar to healthy native tissue [3]. Such muscle grafts can be obtained either by removal of tissue from the patient, an autograft, which results in morbidity and scar formation at the donor site, or by tissue engineering, however currently there is no FDA approved tissue engineering treatment in the USA. This latter approach provides a successful therapy to treat the same disease by avoiding the risk of donor site morbidity, but it involves generating viable tissue in the laboratory by culturing cells on a biocompatible and resorbable tissue engineering scaffold [3,4]. Previous attempts in tissue engineering for oral and craniofacial reconstruction are mostly associated with bone and cartilage tissue regeneration, with scaffolds fabricated from either metallic biomaterials, such as platinum and other alloys, or synthetic resorbable biomaterials, such as poly(lactide-co-glycolide) and polycaprolactone. There is no FDA approved treatment, as of now, for surgical procedures involving biological grafts for tissue engineering in muscle regeneration [5,6,7].

Facial skeletal muscle enables and controls precise movements of parts of the human anatomy [8]. Thus, a scaffold used for skeletal muscle regeneration needs to have a higher extension ability and flexibility compared to that used for bone tissue regeneration. A polymeric biomaterial fabricated in a textile-based tissue engineering structure provides several advantages including flexibility and the ability to engineer a structure with the desired physical and mechanical properties [9]. With the incorporation of biodegradable, bioresorbable polymers, tissue engineering also benefits from avoiding donor site morbidity and the formation of scar tissue, and having no requirement for repeat surgery involving the removal of the biomaterial after complete healing of the tissue [9,10]. An earlier approach of designing a scaffold for skeletal muscle regeneration included fabrication of a phosphate glass fiber-collagen composite scaffold to evaluate its ability to provide in vitro assistance for engineering and regeneration of craniofacial skeletal muscle [9,11]. However, due to the inherent rigidity of phosphate glass fiber, it is challenging to manipulate the fibers into a curved shape of tissue inside the human body. Furthermore, mechanical stimuli of the sharp end of fibers could cause potential severe body immune reaction.

Textile structures are often used as scaffolds in various tissue engineering applications and medical device designs [12,13,14]. This is because textile structures offer a unique combination of desirable properties including flexibility, thickness, strength and elasticity, and bioresorbability when fabricated from biodegradable polymers [10]. Knitted textiles in particular are advantageous over other conventional textile structures, such as woven fabrics and nonwovens, as they provide the added benefit of significantly higher total porosity and a higher surface area for cells to attach to, migrate and proliferate, along with higher stretch and elastic recovery obtained from the interlooped yarn structure [10]. A number of different biodegradable polymers are available, such as polyglycolic acid, polylactic acid, polycaprolactone, polyhydroxyalkanoates and polyurethanes, which can be spun into yarns and then fabricated into textile structures. Each polymer comes with a different set of physical and mechanical characteristics which can provide the unique engineered properties of scaffolds, such as crystallinity, strength, elasticity and resorption rate [10].

The Poisson’s ratio of a material refers to the changes in lateral strain compared to longitudinal strain when a tensile load is applied in one direction. Most flexible materials, including conventional textile structures, shrink in the lateral direction when stretched in the longitudinal direction, exhibiting a positive Poisson’s ratio. On the other hand, auxetic structures are characterized by a negative Poisson’s ratio (NPR), as they expand laterally when stretched in the longitudinal direction and collapse laterally when compressed longitudinally [15]. By demonstrating NPR, a material incorporates some exceptional properties, such as excellent dimensional stability, high shear modulus, and a high capacity to absorb energy [16]. To capitalize on these properties different NPR textile structures have been developed including fibers, yarns, woven and knitted fabrics from different types of geometries [15,16,17]. Knitting has been found to be a superior fabrication method for NPR fabrics because of the variability in knitted structure and design potential [16].

The ultimate goal of this project is to fabricate a biocompatible and bioresorbable scaffold for skeletal muscle to be applied in facial muscle reconstruction surgery in order to treat craniofacial microsomia. In order to reach this goal, this study achieved three objectives. The primary objective was to design a highly porous, dimensionally stable textile scaffold that could provide up to 90 percent total porosity. Auxetic elements were utilized in the design of the scaffold structures to achieve the desired dimensional stability. The second objective was to design and fabricate a scaffold with equivalent strength and elasticity to native skeletal muscle tissue. The third objective was to demonstrate the biocompatibility of the scaffold in terms of cell metabolic activity, cell attachment and proliferation, when cultured with neonatal human dermal fibroblasts during seven days of cell culture. The eventual objective will be to use a dynamic bioreactor for mechanical stimulation of the muscle precursor cells and to regenerate muscle tissue in vitro. In the dynamic bioreactor, the scaffold would not be expected to change its lateral dimension when cyclic loading was applied in the longitudinal direction. In this way the cells residing within the scaffold would only be exposed to strains in the longitudinal direction. To further incorporate more flexibility in terms of bending and torsional properties into the scaffold design, the present study provides an alternative approach for the fabrication and evaluation of a resorbable textile structure that will provide the required physical, mechanical and biological properties for a tissue engineering scaffold that will regenerate craniofacial skeletal muscle.

## 2. Materials and Methods

### 2.1. Scaffold Fabrication

#### 2.1.1. Raw Material

Two scaffolds, namely PCL A and PCL B, were fabricated by using bioresorbable 100% poly-caprolactone multifilament yarn obtained from Guangdong Zhuhai Adhesive Products Co., Ltd., (Zhuhai, Guangdong, China). Those two knitted structures were selected so as to provide the scaffold with dimensional stability and superior stretch and recovery properties when exposed to cyclic loading and unloading in a dynamic bioreactor for in vitro skeletal muscle regeneration. The semi-crystalline elastomeric polymer, polycaprolactone, was selected to further support and enhance the elasticity of the fabric. The basic characteristics of this polycaprolactone multifilament yarn are shown in Table 1.

#### 2.1.2. Knitting of Scaffold Fabrics

The two weft rib structures were knitted on an 18 gauge Shima Seiki Model SIR 123 flat-bed weft knitting machine using a single double arrowhead re-entrant auxetic geometry. The knitting design used for PCL A had a repeat unit size of 40 × 18 stitches, while that for PCL B had a repeat unit size of 20 × 18 stitches, as shown in Figure 1.

#### 2.1.3. Cleaning and Heat Setting

After knitting, both the fabrics were washed with 1 g per liter of non-ionic detergent Triton X-100 in distilled water in an ultrasonic bath and rinsed twice in distilled water to ensure removal of residual detergent. The fabrics were then dried on a pin tenter frame in a Mathis Model THN oven at 30 degrees Celsius for 3 min and heat set at 45 degrees Celsius for 20 s dwell time.

### 2.2. Physical Characterization

#### 2.2.1. Porosity

To estimate the amount of available void space within the scaffolds, their total porosity was measured using the following standard Equation (1):Porosity P = 100 [A × T − W/D]/A × T(1)
where,
A = Area of the fabric in square centimetersT = Thickness of the fabric in centimetersW = Weight of the fabric in grams, andD = Density of the fiber in g/cm^3^

The thickness of the scaffolds was measured in their relaxed state according to the ASTM D1777 “Standard Test Methods for Thickness of Textile Materials”. The standard density for PCL fibers [18] was 1.145 g per cubic centimeter.

#### 2.2.2. Pore Size Distribution

The pore size distribution was determined from SEM photomicrograph images of uncoated scaffold samples taken with a Hitachi Model S-3200 N variable pressure scanning electron microscope (Hitachi, Krefeld, Germany). The images were captured at 30 Pa pressure in a nitrogen atmosphere with an accelerating voltage of 20 kV at magnifications in the 50×–100× range. It was difficult to determine a standard shape for the pores in a highly stretchable knitted fabric. Thus, to standardize the procedure, all the pore size dimensions were measured in the lengthwise vertical direction and a statistical distribution was obtained.

#### 2.2.3. Dimensional Stability

To demonstrate the dimensional stability of the scaffolds, the scaffolds were held between two clamps and manually stretched in the lengthwise direction. The maximum percent extension in the cross direction was measured manually while the specimens were being extended longitudinally. The raveling tendency of the scaffolds was determined visually by observing any fraying or unraveling of loops from the edges of the scaffolds while being extended and relaxed manually between the two clamps.

### 2.3. Mechanical Characterization

#### Bursting Strength and Elongation at Break

The ultimate bursting strength and elongation at break were evaluated according to the standard ISO 7198:2016, “Cardiovascular Implants and Extracorporeal Systems”. A compression cage assembly was mounted on an Instron tensile tester, and each specimen was placed between two horizontal clamping plates. A puncture probe measuring 6 mm in diameter with a smooth spherical end was used to burst the scaffold specimens held horizontally. The load-extension curves were obtained with a 2 kN load cell and traverse speed of 300 mm per minute. We calculated the bursting strength in kPa and the percent elongation at break using the geometry described in Figure 2.

According to the geometry, to calculate the linear elongation at break, the original length was considered to be 12 mm, which was the diameter of the open area where the specimen was held between the clamping plates. The extended length for each specimen was calculated from the Equation (2):Extended length = 2x + π r(2)
where r = 3 mm and x was the hypotenuse of the triangle ABC in Figure 2, where the vertical distance y was read from the displacement measured during the test.

### 2.4. Cell Metabolic Activity and Biocompatibility

#### 2.4.1. Sample Preparation

Scaffold specimens having a circular disc shape of 13 mm in diameter were punched out of the fabrics to fit into 24-well culture plates. Circular glass coverslips measuring 12 mm in diameter were used as the reference control surface as they are known to favor cell attachment and proliferation.

#### 2.4.2. Sterilization of the Scaffolds

The samples were sterilized by exposure to ethylene oxide gas for 12 h at ambient temperature, in an Anprolene Model AN74ix sterilizer (Anderson Products, Inc., Haw River, NC, USA). The samples were then aerated by exposure to air for 48 h so as to evaporate any residual ethylene oxide.

#### 2.4.3. Cell Seeding and Culture

A 7-day in vitro cell culture study was undertaken using both scaffold samples and the glass coverslips for comparison against the reference material. The cell line used contained male neonatal human dermal fibroblasts (Lonza, Inc., Walkersville, MD, USA), expanded to passage 3 and harvested for seeding. On Day 0, 20,000 cells were seeded in each well. The growth medium consisted of Dulbecco’s Modified Eagle Medium (high glucose) (Gibco, Inc., Grand Island, NY, USA) with 10% fetal bovine serum (Specialty Media, Billerica, MA, USA) and 1% antibiotic antimycotic solution (Sigma Life Science, Jerusalem, Israel). All three samples were transferred to empty wells on Day 1, so that the fluorescence measured by the alamarBlue^®^ assay would be derived from only those cells that were attached to the scaffolds, and not from the cells that settled down to the bottom of the well. In addition, an equal number of cells were seeded in an empty well with no scaffold to serve as a positive control for the alamarBlue^®^ assay.

#### 2.4.4. Cell Metabolic Activity Using alamarBlue^®^ Assay

The alamarBlue^®^ assay was used to assess the level of cell metabolic activity and proliferation without damaging the scaffolds or the living cells. The ready-to-use, non-toxic alamarBlue^®^ cell viability reagent from Invitrogen (Thermo Fisher Scientific, Eugene, OR, USA) was used to evaluate the cell metabolic activity on the scaffolds at Day 3 and Day 7 according to the standard protocol recommended by the manufacturer. A media sample with no scaffold and no cells was used as the negative control, and the cells seeded directly in the culture well with no scaffold was used as the positive control. The fluorescence was measured using Gen5 software on a Biotek Model Synergy HT multi-mode microplate reader at excitation/emission wavelengths of 540/590 nm. The % reduction in each sample was calculated according to the following Equation (3):(3)$$\%\;\mathrm{Reduction}\;\mathrm{in}\;\mathrm{sample}=\frac{sample\;fl.\;value\;-\;negative\;fl.\;value}{positive\;fl.\;value\;-\;negative\;fl.\;value}\times 100$$
where,
*Sample fl. value* = fluorescence value obtained from scaffold sample*Negative fl. value* = fluorescence value obtained from negative control (media only)*Positive fl. value* = fluorescence value obtained from positive control (cells only, no biomaterial)

#### 2.4.5. Cell Viability Using Live/Dead Staining Assay and Confocal Microscopy

A live/dead staining assay was carried out for the assessment of cell viability and visualization of cell attachment at Day 3 and Day 7. The live/dead viability/cytotoxicity kit from Invitrogen (Thermo Fisher Scientific, Oregon) uses two dyes, Calcein AM (green) which stains the cytoplasm of live cells, and EthD-1 (red) which stains the nuclei of dead cells. The images of the fluorescent cells were obtained on a Zeiss Model LSM 880 confocal microscope with Zen Black Software at 488 nm and 561 nm excitation wavelengths for Calcein AM and EthD-1 respectively.

### 2.5. Statistical Analysis

Differences between the two scaffold samples were determined statistically using a two-sample *t*-test. Differences in mean values were found to be statistically significant when the *p* values were greater than 0.05.

## 3. Results and Discussion

Two scaffold samples, PCL A and PCL B, were fabricated from poly(ε-caprolactone) multifilament yarn (160 denier, 36 filaments, single filament diameter approximately 20 microns) using weft knitting designs, having geometry in two repeat sizes. These samples were evaluated for their suitability in the regeneration of facial skeletal muscle tissue. The evaluation involved physical and mechanical characterization as well as biological tests in order to determine the performance level of both the samples as well as a comparison between the two.

### 3.1. Characterization of Physical Properties of Knitted Scaffolds

Table 2 includes an overview of the different physical characteristics of both scaffold samples, PCL A and PCL B. PCL A consisted of a 40 × 18 stitch repeat unit, while PCL B consisted of a 20 × 18 repeat unit. The larger repeat unit size, PCL A, reflects a more open knitted structure which results in a lower fabric thickness and weight, and a higher total porosity and average pore size. The tighter knit design of PCL B was observed in its physical properties with higher wales per centimeter and courses per centimeter. Also, PCL B was 1 mm thicker than PCL A, and its fabric weight in grams per square meter was almost twice that of PCL A. Although the pore size range in PCL A was wider than that in PCL B, there was no significant difference in calculated total porosity between the two scaffolds.

The SEM images of both the scaffolds used for typical pore size measurements are shown in Figure 3. Pore sizes ranging from 48 to 846 microns without a common shape or topology were observed to be randomly distributed throughout the fabric specimens. When the two samples were compared, PCL A was found to have a significantly (*p* < 0.05) larger average pore size at 319 microns than PCL B whose average pore size was only 250 microns.

### 3.2. Characterization of Mechanical Properties of Scaffolds

In order to be used as a scaffold for tissue regeneration, the structure needs to be strong enough to sustain a marginally higher load than the original tissue it is replacing. Hence, it was important to evaluate the mechanical properties of the scaffolds, such as the ultimate bursting strength and the elongation at break.

The maximum breaking load and maximum extension at break were determined using the compression cage assembly on the Instron mechanical tester. The scaffold specimen was pre-tensioned, so the crimps were removed, and the fabric was in a flat condition prior to placing it between the horizontal clamping plates. This enabled the relative bursting strengths of the two scaffold samples to be compared with each other, and with the natural skeletal muscle tissue of mammals. According to the geometry, the area under load was taken to be 113.04 mm^2^ and the mean bursting strength values in kPa were calculated from the average maximum load at break for each of the two samples. The bursting strength of a tissue engineering scaffold is expected to be marginally higher than that of the original tissue it will replace. The bursting strength of mammal skeletal muscle tissue is approximately 1075 kPa, while the biaxial elongation at break is approximately 65% [19]. The two PCL fabric scaffold samples gave comparable values for average bursting strength, and PCL A was found to have significantly higher strength than PCL B, as shown in Table 3. (*p* < 0.05).

The percent elongation at break values were compared to that of a mammal skeletal muscle tissue, which has been reported to be 65% under biaxial loading [19]. The elongation at break values for both the scaffolds were found to exceed this reference value. The difference between the percent elongation at break for the two samples was not statistically significant (*p* > 0.05).

Figure 4 shows the load-elongation curves obtained from the two scaffold samples during bursting strength testing compared to that of the abdominal skeletal muscle of a mammal. The initial shape of the curves up to 20 percent elongation at low stress are similar for both the reference tissue and the scaffolds. Figure 4 also shows that the two PCL fabrics are initially able to mechanically support and reinforce the native muscle tissue because of their superior mechanical strength. Though the native tissue is less rigid and more compliant than the scaffolds, the initial curve up to 20% extension is comparable.

### 3.3. Dimensional Stability Determination

Figure 5 shows the changes in dimensions of the scaffolds during manual lengthwise extension. Table 4 compares the change in dimensions and volume of the two scaffolds before and after extension in order to determine their dimensional stability in the lengthwise direction. We observed that both scaffolds showed a high extension in length before any crosswise shrinkage was observed in the width. Also, PCL A showed a 61.8% increase in volume at 76% extension, while PCL B showed an increase of 8.4% in volume at more than 150% extension. This confirms that both scaffolds had a negative Poisson’s ratio during more than 50% extension, which ensures that neither scaffold experienced lateral shrinking during initial extension, which is different from most conventional fabric structures.

Both scaffolds were also evaluated for their tendency to ravel during 100% extension in both the warp and weft directions. Conventional weft knitted structures are easily raveled from the open end when under extension, but auxetic designed fabrics do not have a tendency to ravel, which is an important property for any scaffold that is exposed to a dynamic bioreactor with cyclic loading and unloading.

### 3.4. Cell Metabolic Activity and Biocompatibility

The biocompatibility of the scaffolds was determined using two in vitro cell culture assays; the alamarBlue^®^ assay for cell metabolic activity, and the live/dead staining assay using confocal microscopy to assess cell proliferation and attachment at Day 3 and Day 7.

#### 3.4.1. AlamarBlue^®^ Assay

The results obtained from the alamarBlue^®^ assay plate reading on Day 3 and Day 7 are shown in Table 5 and Figure 6. The percent reduction of alamarBlue^®^ dye for each of the three biomaterials, including two scaffolds and the glass coverslip for reference, was calculated using the Equation (3), given in Section 2.4.4.

The percent reduction is the quantitative measure of the cell metabolic activity on the scaffolds compared to the positive control, as it is directly proportional to the number of live cells in the well. Table 5 summarizes the percent reduction in the scaffold samples PCL A and PCL B and glass coverslip at Day 3 and Day 7. It also compares the percent increase in the cell metabolic activity from Day 3 to Day 7, which shows that although the initial cell attachment on the scaffolds was less compared to the glass coverslip, the increase in cell numbers from Day 3 to Day 7 was more rapid in both scaffold samples compared to the glass coverslip. This indicates that the scaffold samples do promote cell proliferation, but the scaffold surface needs to be improved so as to encourage greater initial cell attachment.

Figure 6 compares the increasing rate of cell metabolic activity in terms of the fluorescence values for all three biomaterials with that of the positive and negative controls. The image supports the claim that the scaffolds encouraged cell proliferation between Day 3 and Day 7, but when compared to the positive control and to the glass coverslips, the scaffolds held a lot of scope for improvement in cell adhesion.

#### 3.4.2. Confocal Microscopy with Live/Dead Staining Assay

Figure 7 describes the comparison of cell attachment on and within the three-dimensional PCL. We attempted to visualize the proportion of viable cells to dead cells and the migration of cells through the scaffolds between Day 3 and Day 7. It can be seen from Figure 7a that the cells attached to the individual filament surfaces favored residing in the space between the filaments and not in the pores between yarns. Figure 7b shows that the cells penetrated up to a distance of 450 microns deep into the three-dimensional scaffold, both at Day 3 and Day 7. This shows that the morphology and structure of the scaffolds helped the cells to be distributed up to 450 microns into the scaffolds, but no deeper. The depth of cell penetration could have been influenced by the initial cell seeding density or by differences in the scaffold structure that affects cell seeding or migration. Figure 7c,d visually compares the proportion of live and dead cells on the scaffold sample PCL A between Day 3 and Day 7. The live cells were observed to be well attached and elongated on Day 7, but there was no difference between Day 3 and Day 7 dead cells, which supported the results obtained from the alamarBlue^®^ assay.

Similarly, Figure 8 describes the morphology and compares the proportion and distribution of live and dead cells on scaffold sample PCL B. Please note that there was no noticeable difference observed between the proportion of live and dead cells at Day 3 and Day 7 during the fibroblast cell culture on this sample. However, from the morphology of the live cells, on Day 7 the cells were observed to be healthy compared to Day 3, in terms of the number of elongated cells and the average length of cells. The effect of scaffold pore size may have influenced the appearance and orientation of the cells attached to the filament surface. It was noted that the cells on the PCL A scaffold with its larger average pore size, which would have provided better fluid transfer, were observed on Day 7 to be more intimately attached compared to those on the PCL B scaffold with smaller pores. When viewing the three-dimensional z-stack image of the scaffold, the depth of cell distribution was observed to be about 450 microns on Day 3 and about 400 microns on Day 7. The significance of the cell penetration results would have been easier to understand if the culture study had been extended further up to 14 and 28 days.

## 4. Conclusions and Future Work

The poly(ε-caprolactone) multifilament yarn was successfully fabricated into two knitted textile scaffolds, PCL A and PCL B. By selecting an open weft knit structure for both samples, total porosities in excess of 90 percent were achieved. While neither sample showed a tendency to ravel, PCL A showed superior dimensional stability compared to PCL B. The results of bursting strength and biaxial elongation at break were compared with reference values obtained from skeletal muscle tissue reported in the literature. The load-elongation curves and initial stiffness values of both scaffold samples were compared to those of the reference skeletal muscle tissue, and the ultimate strength of the scaffolds exceeded that of the native tissue, thereby ensuring mechanical support and reinforcement during the initial period after implantation. The biological performance of the scaffolds was measured in terms of alamarBlue^®^ and live/dead assays using confocal microscopy. This has enabled an evaluation of the metabolic activity, attachment and distribution of neonatal human dermal fibroblast cells on both scaffolds. The PCL Fabric A was found to support cell growth better than PCL B in terms of cell metabolic activity after 7 days of cell culture.

Since the scaffolds are intended for the regeneration of facial skeletal muscle, the next step will be to evaluate the biocompatibility with muscle precursor cells in vitro and to determine the ability to regenerate muscle tissue in vivo. The resorption rate of the scaffolds needs to be evaluated to compare it with the time period for complete healing and growth of the original tissue in situ. The capacity of the knitted structures to provide dimensional stability during mechanical stimulation of the scaffold must also be evaluated by culturing the skeletal muscle cells on the scaffolds in a dynamic bioreactor.

## Figures and Tables

**Figure 1 bioengineering-07-00134-f001:**
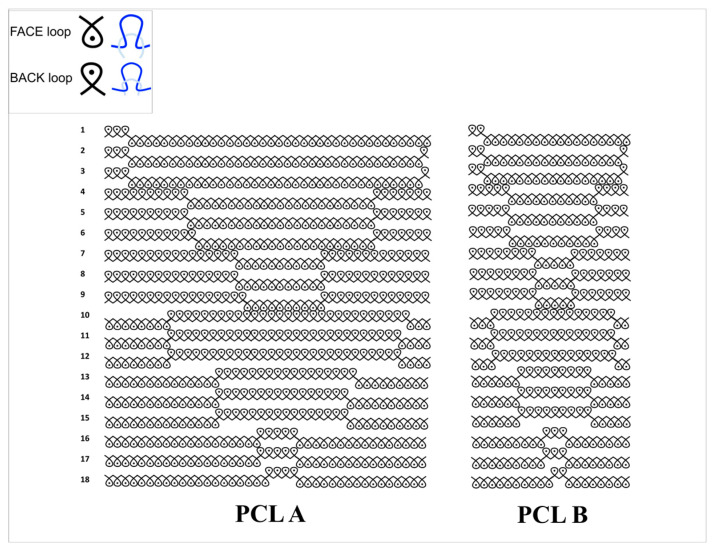
Knitting design notations of scaffold samples PCL A (**left**) and PCL B (**right**). The inset on the top left explains the actual appearance of the components in the design notations in terms of face loop and back loop.

**Figure 2 bioengineering-07-00134-f002:**
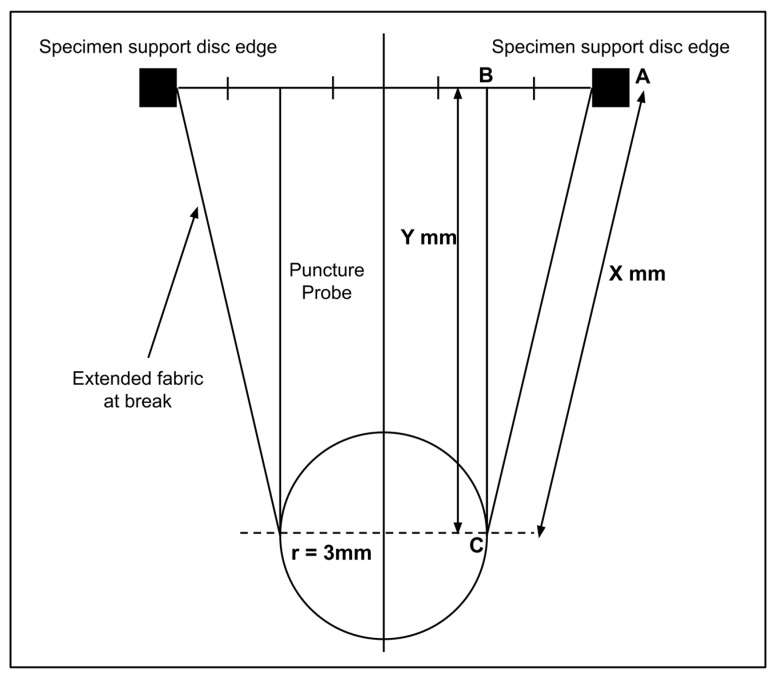
The geometry for determining the specimen area and sample extension when measuring the ultimate bursting strength and elongation at break of samples punctured in the compression cage according to the Standard ISO 7198:2016, “Cardiovascular Implants and Extracorporeal Systems.”.

**Figure 3 bioengineering-07-00134-f003:**
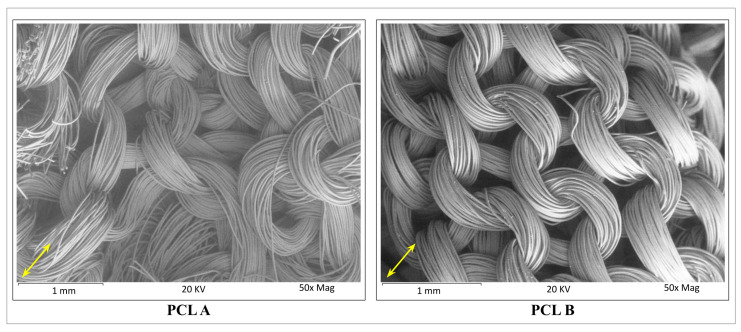
Scanning electron microscope images of scaffolds **PCL A** and **PCL B**. The average pore size was determined from the width of each pore always measured in the same direction. A total of 50 pore size dimensions from five different images of each scaffold sample were measured and averaged. The arrows show the direction of measurement for each of the SEM images.

**Figure 4 bioengineering-07-00134-f004:**
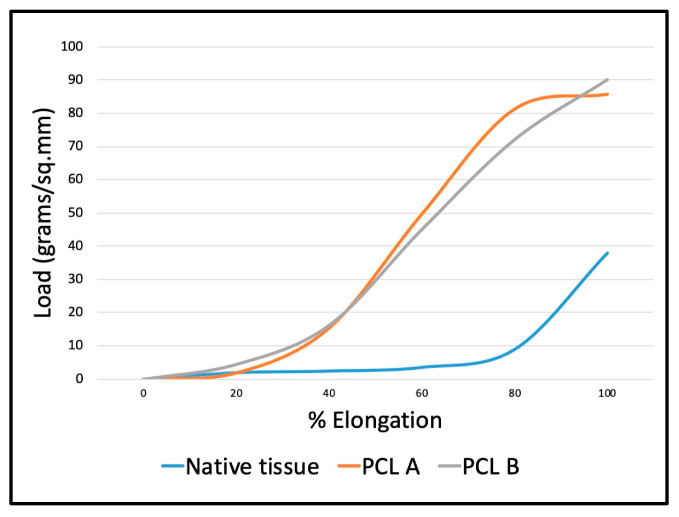
Comparison of load-elongation curves for the scaffolds PCL A and PCL B obtained from the probe puncture test compared to that of the abdominal skeletal muscle of mammals.

**Figure 5 bioengineering-07-00134-f005:**
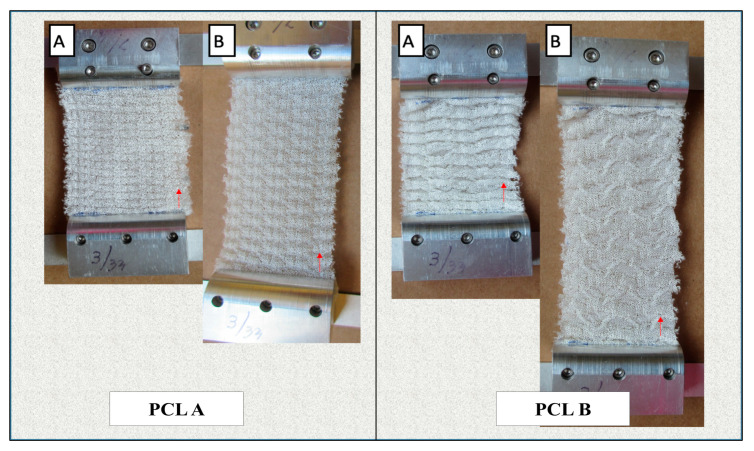
Dimensional behavior of the scaffolds PCL A and PCL B. Both scaffolds were held and stretched between two clamps vertically to show the scaffolds did not contract in widthwise direction. Figure A: Before tension was applied. Figure B: After tension was applied.

**Figure 6 bioengineering-07-00134-f006:**
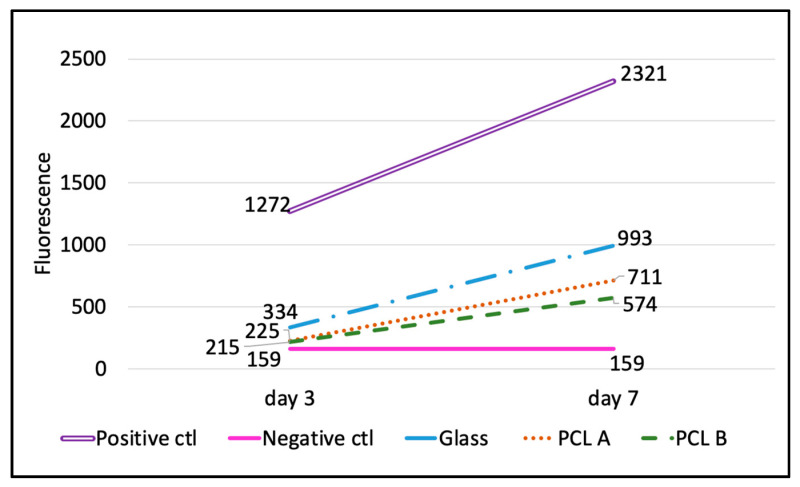
Level of cell metabolic activity in terms of fluorescence values at Day 3 and Day 7 during cell culture. The level of cell metabolic activity of fibroblasts cultured on glass coverslips and on PCL A and PCL B scaffolds were compared between each other. Only growth media with no cells or biomaterials was used as the negative control and fibroblasts cultured directly over the culture well-plate were used as the positive control so as to identify the range of fluorescence.

**Figure 7 bioengineering-07-00134-f007:**
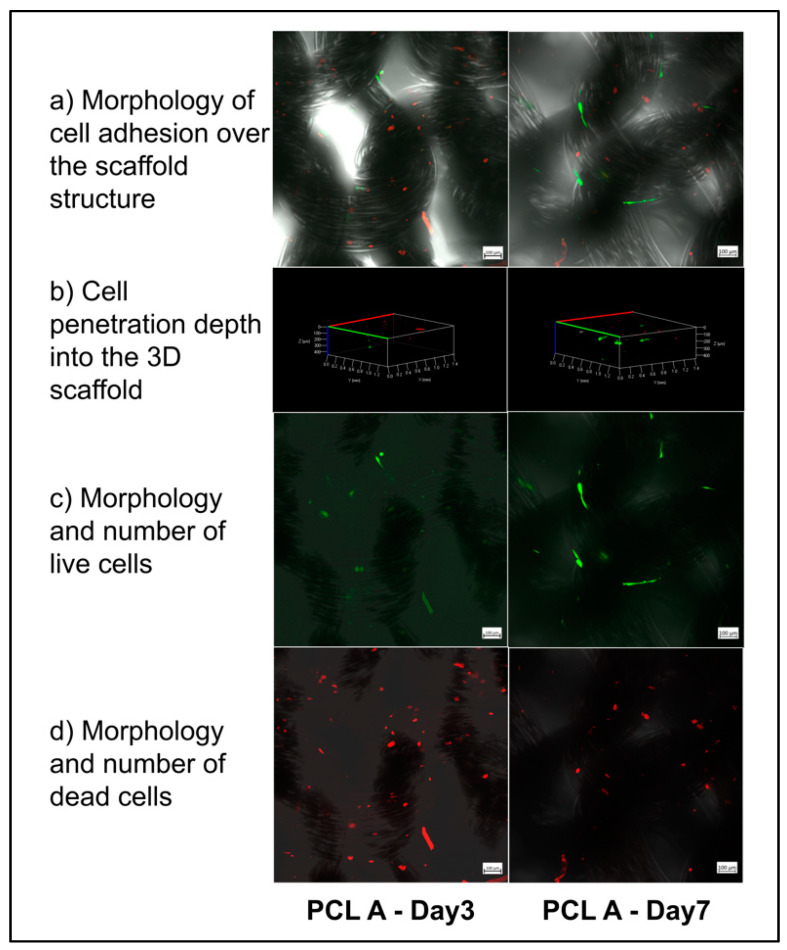
Confocal imaging of fibroblast culture on the PCL A scaffold at Day 3 and Day 7. The live cells were stained with Calcein-AM (green) and the dead cells were stained with EthD-1 (red) fluorescent dyes.

**Figure 8 bioengineering-07-00134-f008:**
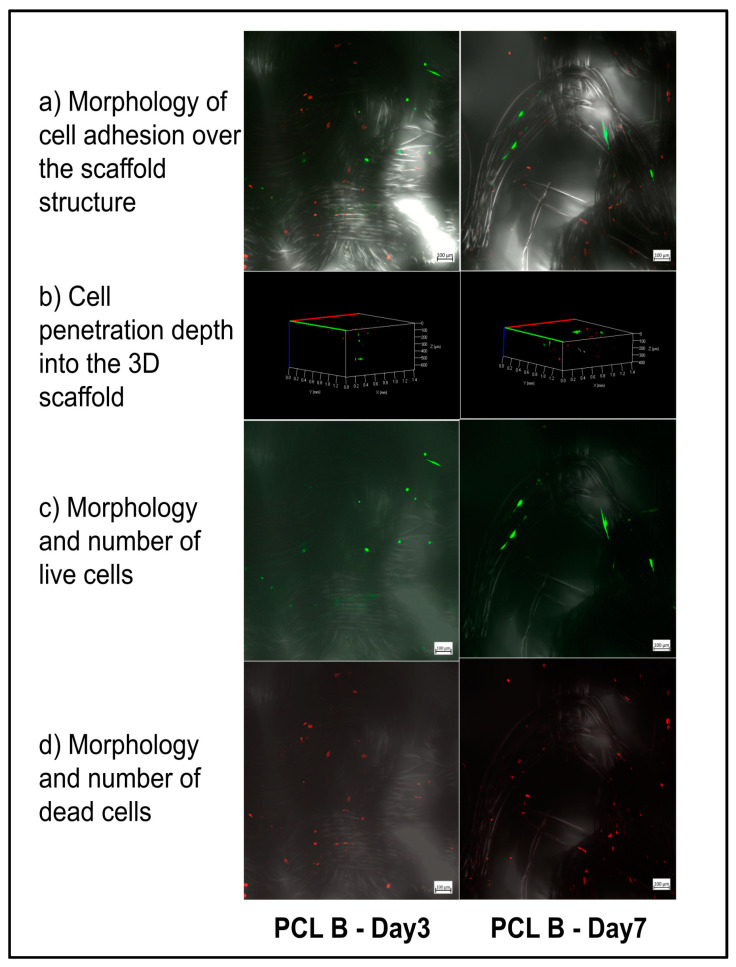
Confocal imaging of fibroblast culture on the PCL B scaffold at Day 3 and Day 7. The live cells were stained with Calcein-AM (green) and the dead cells were stained with EthD-1 (red) fluorescent dyes.

**Table 1 bioengineering-07-00134-t001:** Physical properties of polycaprolactone yarn. The glass transition temperature and melting temperature show that the polymer is in a semi-crystalline state at room temperature which explains the high elongation at break and its suitability for this application.

Polymer	Approx. Average Mw	Denier	Filaments	Elongation at Break at RT	Tg	Tm	Cross Section	Storage Condition
Poly-caprolactone	80 kDa *	160	36	>150%	−60 °C	60 °C	Circular	4 °C

* The information is as provided by the manufacturer.

**Table 2 bioengineering-07-00134-t002:** Design parameters and physical characteristics of scaffolds PCL A and PCL B.

Scaffold Sample	Repeat Unit Size	No. of Courses/cm	No. of Wales/cm	Average Fabric Thickness (mm)	Fabric Weight (g/m^2^)	Total Porosity (%)	Pore Size Range (µm)	Average Pore Size (µm)
PCL A	40 × 20	9	34	0.87 ± 0.11	259	95	48–846	319
PCL B	20 × 20	14	44	1.84 ± 0.07	436	94	86–595	250

**Table 3 bioengineering-07-00134-t003:** Comparison of average bursting strength and average elongation at break of scaffolds PCL A and PCL B with reference values of mammalian skeletal muscle tissue.

Scaffold Samples	Average Bursting Strength (kPa)	Average Elongation at Break (%)
Native tissue	1075	65
PCL A	900 (±46)	293 (±50)
PCL B	856 (±48)	318 (±36)

**Table 4 bioengineering-07-00134-t004:** Changes in physical dimensions on stretching in the warp direction for samples PCL A and PCL B. The scaffolds showed an increase in volume when the samples were stretched, demonstrating a negative Poisson’s ratio.

Scaffold Samples	Before Extension in Warp Direction				After Extension in Warp Direction				Change in Volume
	Length (mm)	Width (mm)	Thickness (mm)	Volume (mm^3^)	Length (mm)	Width (mm)	Thickness (mm)	Volume (mm^3^)	
PCL A	50	50	0.87	2175	88	50	0.8	3520	+61.8%
PCL B	40	50	1.84	3680	105	50	0.76	3990	+8.4%

**Table 5 bioengineering-07-00134-t005:** The percent reduction in the fluorescence of alamarBlue^®^ dye for scaffolds PCL A and PCL B exposed to fibroblast cell culture on Day 3 and Day 7 and the percent increase in cell metabolic activity compared between the two scaffold samples and with the glass coverslip.

Scaffold Samples	% Reduction		% Increase in Cell Metabolic Activity from Day 3 to Day 7
	Day 3	Day 7	
Glass	16 (±10)	49 (±7)	210
PCL A	6.0 (±1.0)	26 (±4)	330
PCL B	5.0 (±1.0)	19 (±3)	280

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
