# Peer review of "Poly(ε-Caprolactone) Resorbable Auxetic Designed Knitted Scaffolds for Craniofacial Skeletal Muscle Regeneration"

_bioengineering, 2020, doi:10.3390/bioengineering7040134_

Round 1
Reviewer 1 Report
The article entitled " Poly(ε-caprolactone) Resorbable Auxetic Designed Knitted Scaffolds for Craniofacial Skeletal Muscle Regeneration" addresses current problem of autograft muscle treatment, proposing tissue engineering based concept, here specifically for craniofacial skeletal muscle regeneration. The general characterization of obtained scaffolds and cell cultured studies are correctly presented and proving the usability of the developed systems, nevertheless the paper has significant lack of presented mechanical results .
Major comments:
- In section 3.2 Authors presented the mechanical properties of the scaffolds in general and laconic style. Although the maximum breaking load and maximum extension at break were determined no measured values are presented in the paper, the same as for bursting strength. The only certain values are for mammal skeletal muscle tissue. Authors need to summarized all calculated values. It is impossible to assess the correctness of Authors statements, e.g.” The elongation at break values for both the scaffolds were found to exceed this reference value. And the difference between the percent elongation at break for the two samples was not statistically significant (p > 0.05).” without the presented, measured values.
- Similar comment concerns the tendency to ravel. On page 10 Authors state that this property was evaluated but no further results or comments are presented.
Minor comments:
- In subsection 3.3. the question mark in title should be probably removed.
- Equations should be renumbered – first is on page 4. Please add equation 2 and 3 to Materials and Methods part.
- Table 4 – the title is too long. Sentences: “The percent reduction was directly proportional to the number of live cells which corresponded to the level of cell metabolic activity. The increase in cell metabolic activity was determined by calculating the percentage difference between the reduction …” are repetition of the comments from the main text.
- Confocal microscope images presented in Fig. 7 and 8 are poor quality – no details are visible. Scale bars (Fig.7c,d, 8c,d) are unreadable, same as three-dimensional models representing cell penetration into the scaffolds. Please increase the contrast on above mentioned images.
Reviewer 2 Report
This paper is developed a cell scaffold with Poisson's ratio by weaving biodegradable polymer PCL fibers.
In the cell scaffold manufacturing method, it is generally a fiber weaving technology, so it will be easier to understand if the following parts are added.
The thickness of the fiber is explained as Denier 160 / Filaments 36, however it would be better to explain the thickness or diameter of the fiber after weaving.
In addition, it would be better to measure the size of the fiber bundle after weaving if it can be measured.
PCL B does not seem to be significantly different from A in the penetration of cells, is it the property of the cell, and is porosity not effective?
Is the cell support not coated for cell adhesion? A coating such as the collagen or something will help in cell adhesion.
Number 4 is missing, '5. Conclusions and Future Work' came after '3.4.2. confocal microscopy with Live/Dead staining assay'.
Round 2
Reviewer 1 Report
I accept all introduced changes.